# A Data-Weighted Prior Estimator for Forecast Combination

**DOI:** 10.3390/e21040429

**Published:** 2019-04-23

**Authors:** Esteban Fernández-Vázquez, Blanca Moreno, Geoffrey J.D. Hewings

**Affiliations:** 1REGIOlab and Department of Applied Economics, University of Oviedo, Faculty of Economics and Business, Avda. del Cristo, s/n, 33006 Oviedo, Spain; 2Regional Economics Applications Laboratory, University of Illinois at Urbana-Champaign 607 S. Matthew, Urbana, IL 61801-367, USA

**Keywords:** data-weighted prior, generalized maximum entropy method, combined forecast

## Abstract

Forecast combination methods reduce the information in a vector of forecasts to a single combined forecast by using a set of combination weights. Although there are several methods, a typical strategy is the use of the simple arithmetic mean to obtain the combined forecast. A priori, the use of this mean could be justified when all the forecasters have had the same performance in the past or when they do not have enough information. In this paper, we explore the possibility of using entropy econometrics as a procedure for combining forecasts that allows to discriminate between bad and good forecasters, even in the situation of little information. With this purpose, the data-weighted prior (DWP) estimator proposed by Golan (2001) is used for forecaster selection and simultaneous parameter estimation in linear statistical models. In particular, we examine the ability of the DWP estimator to effectively select relevant forecasts among all forecasts. We test the accuracy of the proposed model with a simulation exercise and compare its *ex ante* forecasting performance with other methods used to combine forecasts. The obtained results suggest that the proposed method dominates other combining methods, such as equal-weight averages or ordinal least squares methods, among others.

## 1. Introduction

Forecasting agents can use an ample variety of forecasting techniques and different information sets, thus leading to a wide variety of obtained forecasts. Hence, as each individual forecast captures a different aspect of the available information, a combination of them would be expected to perform better than the individual forecasts. In fact, a growing volume of literature has demonstrated that a combined forecast increases forecast accuracy in several fields (e.g., [1,2,3,4,5,6,7]).

The first study about the forecast combination was carried out by [8]. Since their study, several researchers have shown a variety of modeling procedures to estimate the weights of each individual forecast in the combined forecast (a review of the literature can be found in [5,9,10]).

There are several methods for forecast combination that can be classified as variance–covariance methods, probabilistic methods, Bayesian methods, or regression-based methods, among others. The first kind of method allows the calculation of weights of the combined forecast by minimizing the error variance of the combination ([8,11]); Probabilistic methods ([12,13]) weights are linked to the probability that an individual forecast will perform best on the next occasion; Bayesian methods, which were originally put forward by [14], assume that the variable being predicted (*y*) and the individual forecasts have a random character and the combined forecast is the expected value of the a posteriori distribution of *y* that is modified from its a priori distribution with the sample information of the individual forecasts ([14,15,16,17,18], among others).

The regression-based methods were introduced by [19]. These methods link the weights of the combined forecasts to the coefficient vector of a linear regression, where individual forecasts are explanatory variables of the variable being predicted. The estimation of the coefficient vector is based on the past available information of individual forecasts and realizations of the variable being predicted. However, when the number of agents providing forecasts increases, the combined regression method involves the estimation of a large number of parameters and a dimensionality problem could arise.

In such a situation, in order to take out relevant information from a large number of forecasts, some procedures can be used, such as the subset selection, factor-based methods ([20,21]), ridge regression [22], shrinkage methods [23], latent root regression [24] or least absolute shrinkage, and the selection operator method ([25,26]), among others. Nevertheless, the simple arithmetic mean of the individual forecasts is the most used strategy to obtain the combined forecast. This strategy could be justified, as some researchers have empirically shown that simple averaging procedures dominate other, more complicated schemes ([2,27,28,29], among others). Such a phenomenon is usually referred to as the “forecasting combination puzzle” which has been documented by [10], who shows that the simple arithmetic mean constitutes a benchmark. From a theoretical point of view, the simple equal-weight average could be justified when all the forecasters have shown the same forecast performance in the past, or there is not available information about individual forecast´s past performance to calibrate them differently.

In such a situation of limited information, the following question arises: Could it be possible to combine individual forecasts differently from the simple average procedure? This drawback of the combination forecast is one of the potential problems which we address in this paper. In fact, under a regression-based combination method framework we propose a procedure that allows for simultaneous parameter estimation and forecast selection in linear statistical models. This procedure is based on the data-weighted prior (DWP) estimator proposed by [30]. This estimator has been previously applied to standard regression analysis, but not specifically to the field of forecast combination. More specifically, we analyze how DWP is able to reduce the number of potential forecasters and estimate a vector of weights different from the simple average in the combined forecast. We use a simulation exercise to compare the ex-ante forecasting performance of the proposed method with other combining methods, such as equal-weight averages or ordinal least square methods, among others. The obtained results indicate that the method based on DWP outperform other examined forecast combination methods.

The paper is organized in five additional sections. Section 2 introduces the framework of the regression-based combination methods. Section 3 presents the data-weighted prior (DWP) estimator. Section 4 shows the simulation experiment and presents the results. Finally, Section 5 summarizes the conclusions of the research.

## 2. Forecast Combination Methods Based on Regression Methods

There is a large number of individual forecasts to forecast any given variable (*y*) with forecast horizon *h* at time *t*, $y_{t+h}$. We indicate by $x_{it}$ the forecast referred to t+h, given in period t by a forecasting agent or model i (i=1,…, K). The theory of combining forecasts indicates that it could be possible to obtain an aggregated prediction ${\hat{y}}_{t}$ that combines the individual forecasts $x=\left(x_{1t},\ldots ,x_{Kt}\right)$ through a vector of weights $\beta ={\left(\beta_{1},\ldots ,\beta_{K}\right)}^{\prime }$.

The first study about forecast combination focused on the combination of two forecasts whose vector of weights was obtained from the error variances of the individual forecast [8]. Afterward, [11] showed a combined forecast obtained by ${\hat{y}}_{t}=x\beta$, with the sum of weights is $l^{\prime }\beta =1$, l being a vector (K×1) of ones and $0\leq \beta_{i}\leq 1$. The combined forecast reduces its error variance since:(1)$$\hat{\beta }=\frac{\left(\Sigma^{-1}l\right)}{\left(l^{\prime }\Sigma^{-1}l\right)};\;\mathrm{where}\sum^{\;}=E\left(e_{t}e_{t}^{\prime }\right)\;\mathrm{and}\;e_{t}=yl^{\prime }-x,$$
where $e_{t}$ is the vector (K×1) containing the forecast error specific to each forecasting agent or model i.

However, the method does not take into account the possible correlation in the errors of the forecasts being combined. [19] showed that weights of the combined forecasts obtained through conventional methods can be interpreted as the coefficient vector of the linear projection of the variable being predicted from the K individual forecasts as:(2)$$y_{t+h}=x\beta +e_{t+h},$$
where $y_{t+h}$ is the variable being predicted (unobservable). The estimation of β is based on the past observations of the variable $y=\left(y_{1},\;y_{2},\ldots ,\;y_{T}\right)$ and experts’ past performances $X=\left(x_{1},\ldots ,x_{K}\right)$:(3)y=Xβ+ϵ,
where y is a (T×1) vector of observations for y, X is a (T×K) matrix of experts’ past performances, being each $x_{i}$ a *T* × 1 vector of individual past forecasts, β is the (K×1) vector of unknown parameters $\beta =\left(\beta_{1},\ldots ,\beta_{K}\right)$ to be estimated, and ϵ is a (T×1) vector with the random term of the linear model.

The combining regression-based methods introduced by [19] were extended in several ways. Thus, [31] introduced time varying combining weights and [32] introduced nonlinear specifications in combined regression context. The dynamic combined regressions were introduced by [33] to take into account the serially correlated errors. Moreover, [34,35] considered the problem of non-stationarity.

However, the number of institutions carrying out forecasts has increased considerably in the last few years, thus the projection methodology suggested by Equation (3) would involve the estimation of a large number of weights. Thus a “curse of dimensionality problem” could arise when losing degrees of freedom for the regression estimation. In such cases, it is usual to use the simple mean average of the individual forecasts as a combined forecast.

In this situation of limited information about the past performance of individual forecasts, a question that arises is how to combine individual forecasts differently from the simple mean average. Some authors have shown evidence in support of an alternative that allows the calibration of individual forecasts when the small amount of information available does not allow the use of regression procedures. In a context where entry and exit of individual forecasters makes the regression estimation unfeasible, [36] shows how an affine transformation of the uniform weighted forecast performs reasonably well in small samples. [6] proposes a combination method based on the generalized maximum entropy approach [37]. Through the application of the maximum entropy principle, their method leads the adjustment of a priori weights (which are associated with the simple mean average) into posterior weights by considering a large number of forecasters, for which there is limited available information about their past performances.

## 3. A Data-Weighted Prior (DWP) Estimator

Generalized cross entropy (GCE) technique has interesting properties when dealing with ill-conditioned datasets (those affected by significant collinearity or small samples) An extensive description of the entropy estimation approach can be found in [37,38]. Thus, in this section we propose the application of an extension of the GCE technique in the context of combining individual predictors.

Let us suppose we are interested in forecasts of a variable y that depends on K explanatory variables $x_{i}$:(4)y=Xβ+ϵ,
where y is a (T×1) vector of observations for the variable being predicted y, X is a (T×K) matrix of observations for the $x_{i}$ variables, β is the (K×1) vector of unknown parameters to be estimated $\beta ={\left(\beta_{1},\ldots ,\beta_{K}\right)}^{\prime }$, and ϵ is a (T×1) vector containing the random errors. Each unknown parameter $\beta_{i}$ is assumed to be a discrete random variable with M≥2 possible realizations. We suppose that there is some information about those possible realizations based on the researcher’s a priori beliefs about the likely values of $\beta_{i}$. That information is included in a support vector $b^{\prime }=\left(b_{1},\ldots ,b_{M}\right)$ with corresponding probabilities ${p^{\prime }}_{i}=\left(p_{i1},\ldots ,p_{iM}\right)$. Although each parameter could have different *M* values, it is assumed that the *M* values are the same for every parameter. Thus, vector ***β*** can be rewritten as:(5)$$\beta =\left[\begin{array}{c} \beta_{1}\\ \vdots \\ \beta_{K} \end{array}\right]=BP=\left[\begin{array}{cc} \begin{array}{cc} b^{\prime } & 0\\ 0 & b^{\prime } \end{array} & \begin{array}{cc} \cdots & 0\\ \cdots & 0 \end{array}\\ \begin{array}{cc} \vdots & \vdots \\ 0 & 0 \end{array} & \begin{array}{cc} \ddots & \vdots \\ \cdots & b^{\prime } \end{array} \end{array}\right]\left[\begin{array}{c} p_{1}\\ \begin{array}{c} p_{2}\\ \vdots \\ p_{K} \end{array} \end{array}\right],$$
where B and P are matrixes with dimensions (K×KM) and (KM×1) respectively. The following expression gives each parameter $\beta_{i}$ as:(6)$$\beta_{i}=b^{\prime }p_{i}=\overset{M}{\underset{m=1}{\sum }}b_{m}p_{im};\;i=1,\ldots ,K$$

A similar approach is followed for ϵ. It is highlighted that, although GCE does not require rigid assumptions about the probability distribution function of the random error, as with other traditional estimation methods, some assumptions are still necessary to be made. It is assumed that ϵ has a mean E[ϵ]=0 and a finite covariance matrix. Moreover, each element $\epsilon_{t}$ is considered to be a discrete random variable with J≥2 possible values contained in the vector $v^{\prime }=\left\{v_{1},\ldots ,v_{J}\right\}$. Although each $\epsilon_{t}$ could have different *J* values, it is assumed as common for all of them $\epsilon_{t}$ (t=1,…, T). We also assume that the random errors are symmetric around zero ($-v_{1}=v_{J}$). The upper and lower limits ($v_{1}$ and $v_{J},$ respectively) are fixed by applying the three-sigma rule (see [37,38,39]). Thus, vector ϵ can be defined as:(7)$$\epsilon =\left[\begin{array}{c} \epsilon_{1}\\ \vdots \\ \epsilon_{T} \end{array}\right]=VW=\left[\begin{array}{cc} \begin{array}{cc} v^{\prime } & 0\\ 0 & v^{\prime } \end{array} & \begin{array}{cc} \cdots & 0\\ \cdots & 0 \end{array}\\ \begin{array}{cc} \vdots & \vdots \\ 0 & 0 \end{array} & \begin{array}{cc} \ddots & \vdots \\ \cdots & v^{\prime } \end{array} \end{array}\right]$$
and each element $\epsilon_{t}$ has the value equals:(8)$$\epsilon_{t}=v^{\prime }w_{t}=\overset{J}{\underset{j=1}{\sum }}v_{j}w_{tj};\;t=1,\ldots ,T$$

Therefore, model (7) can be transformed into:(9)y=XBP+VW

In this context, we need to estimate the elements of matrix P, but also the elements of matrix W (denoted by ${\tilde{w}}_{tj}$). The problem of the estimation of the vector of unknown parameters $\beta ={\left(\beta_{1},\ldots ,\beta_{K}\right)}^{\prime }$ for the general linear model is transformed into the estimation of K+T probability distributions. Based on this idea, [30] proposed an estimator that simultaneously allows for the estimation of parameters and the selection of variables in linear regression models. In order to have a basis for extraneous variable identification and coefficient reduction, the estimator uses sample but also non-sample information, as it is related to the Bayesian method of moments (BMOM) (see [40,41]). In other words, this technique allows for classifying some the explanatory variables in the linear model as irrelevant by shrinking the coefficients. Recent empirical applications of this method can also be found in [42,43,44].

Focusing on the context of combination of predictions, the objective of the DWP estimator is to identify which individual forecaster should receive a weight significantly different from the equal weighting scheme (simple arithmetic mean) and simultaneously to forecast the target variable based on a combination of individual predictors. We begin by specifying a discrete support space ***b*** for each $\beta_{i}$ symmetric around the value 1/K and with large lower and upper limits, so that each $\beta_{i}$ is contained in the chosen interval with high probability. The upper and lower bounds for ***v*** ($v_{1}$ and $v_{J}$, respectively) are fixed by applying the three-sigma rule. For the estimation of the $\beta_{i}$ parameters, the specification of some a priori distribution ***q*** for the values in the supporting vectors is required. Besides fixing a uniform probability distribution that will be used as ***q*** in the GCE estimation (i.e., $q_{m}=\frac{1}{M}$), we also specify a “spike” prior for each $\beta_{i}$, where a very high probability $q_{m}≅1$ is associated with the value 1/K for $b_{m}$ (i.e., $q_{m}≅0$ for the remaining values). Thus, data-based prior is specified so flexibly that for each $\beta_{i}$ coordinate either a spike prior at the $b_{m}=1/K$, a uniform prior over support space ***b***, or any convex combination of the two, can result. The weight (a weighted formulation in an entropy optimization problem has been also proposed by [45] who proposed a weighted generalized maximum entropy (W-GME) estimator where different weights are assigned to the two entropies (for coefficient distributions and disturbance distributions) in the objective problem. Moreover, under a linear regression model estimation, [46] proposed a streaming generalized cross entropy (Stre-GCE) method to update the estimation of the parameters $\beta_{i}$ by combining prior information and new data) given to the spike prior $q^{s}$ for each parameter $\beta_{i}$ is given by $\gamma_{i}$. For each $\gamma_{i}$, a discrete support space $b_{i}^{\gamma }$ is specified with *n* possible values (n=1,…, N) and corresponding probability distribution $p_{i}^{\gamma }$. Thus, $\gamma_{i}$ is defined as $\gamma_{i}=\sum_{n=1}^{N}b_{in}^{\gamma }p_{in}^{\gamma }$, where $b_{i1}^{\gamma }=0$ and $b_{iN}^{\gamma }=1$ are, respectively, the lower and upper bounds defined as the support of these parameters.

If $q^{u}$ and $q^{s}$ denote the uniform and spike a priori distributions, respectively, we can achieve the objective proposed by minimizing the following constrained problem:(10)$$\begin{array}{ll} \underset{P,P^{\gamma },W}{\mathrm{Min}}D\left(P,P^{\gamma },W\|Q,Q^{\gamma },W^{0}\right)= & \overset{K}{\underset{i=1}{\sum }}\left(1-\gamma_{i}\right)\overset{M}{\underset{m=1}{\sum }}p_{im}ln\left(\frac{p_{im}}{q_{im}^{u}}\right)\\ & +\overset{K}{\underset{i=1}{\sum }}\gamma_{i}\overset{M}{\underset{m=1}{\sum }}p_{im}ln\left(\frac{p_{im}}{q_{im}^{s}}\right)\\ & +\overset{K}{\underset{i=1}{\sum }}\overset{N}{\underset{n=1}{\sum }}p_{in}^{\gamma }ln\left(\frac{p_{in}^{\gamma }}{q_{in}^{\gamma }}\right)\\ & +\overset{T}{\underset{t=1}{\sum }}\overset{J}{\underset{j=1}{\sum }}w_{tj}ln\left(\frac{w_{tj}}{w_{tj}^{0}}\right) \end{array}$$
subject to:(11)$$y_{t}=\overset{K}{\underset{i=1}{\sum }}\overset{M}{\underset{m=1}{\sum }}b_{m}p_{im}x_{it}+\overset{J}{\underset{j=1}{\sum }}v_{j}w_{tj};\;\;t=1,\ldots ,T$$
(12)$$\overset{M}{\underset{m=1}{\sum }}p_{im}=1;\;i=1,\ldots ,K$$
(13)$$\overset{J}{\underset{j=1}{\sum }}w_{tj}=1;\;t=1,\ldots ,T$$
(14)$$\overset{N}{\underset{n=1}{\sum }}p_{in}^{\gamma }=1;\;i=1,\ldots ,K$$
(15)$$\gamma_{i}=\overset{N}{\underset{n=1}{\sum }}b_{in}^{\gamma }p_{in}^{\gamma }$$

The $\gamma_{i}$ parameters and the $\beta_{i}$ coefficients of the model in (10) are estimated simultaneously. Please note the symmetry between the terms γ and 1−γ. Permuting the part of the objective function (10) to which they are connected would not change the final result in terms of the weighting scheme estimated.

To understand the logic of the DWP estimator, an explanation regarding the objective function (10) is useful, which is divided into four terms. The first one measures the divergence between the posterior probabilities and the uniform priors for each $\beta_{i}$ parameter, this being part of the divergence weighted by $\left(1-\gamma_{i}\right)$. The second element of (10) measures the divergence between the uniform priors for each $\beta_{i}$ with the spike prior and it is weighted by $\gamma_{i}$. The third element in (10) relates to the Kullback divergence of the weighting parameters $\gamma_{i}$. It is highlighted that the a priori probability distribution fixed for each one of those parameters is always uniform ($q_{i}^{\gamma }=\frac{1}{N}\;\forall n=1,\ldots ,N$). The last term measures the Kullback divergence between the prior and the posterior probabilities for the random error of the model. The prior distribution of the errors is uniform (again $w_{tj}^{0}=\frac{1}{J}\;\forall t=1,...,T$).

From the recovered ${\tilde{p}}_{im}$ probabilities, the estimated value of each parameter $\beta_{i}$ is obtained as:(16)$${\tilde{\beta }}_{i}=\overset{M}{\underset{m=1}{\sum }}b_{m}{\tilde{p}}_{im};\;i=1,\ldots ,K$$

Under some mild assumptions (see [30], page 177), there is a guarantee that DWP estimates are consistent and asymptotically normal. Moreover, it is also ensured that the approximate variance of the DWP estimator is lower than the approximate variance of the GCE estimator, where the variance is lower than the approximate variance of an Maximum Likelihood- Least Squares estimator (see [30], page 179).

As it was highlighted, the DWP estimator allows simultaneously the estimation of parameters and the selection of predictors in linear regression models. The strategy to reach this objective has two steps. First, the estimates of the weighting parameters $\gamma_{i}$ are obtained as:(17)$${\tilde{\gamma }}_{i}=\overset{N}{\underset{n=1}{\sum }}b_{in}^{\gamma }{\tilde{p}}_{in}^{\gamma };\;i=1,\ldots ,K$$
which can be used as a tool for this purpose: As ${\tilde{\gamma }}_{i}\to 0$**,** the prior gets closer to the uniform and the estimated parameters approach those of the GME estimator. This indicates that the parameter associated with this predictor can take values far from the center of the support vector (i.e., 1/K). On the other hand, for large values of ${\tilde{\gamma }}_{i}$, the part of the objective function with the spike prior on 1/K takes over. Consequently, the predictors considered in the combination that should receive a weight equal to those in a simple mean average will be characterized by large values of ${\tilde{\gamma }}_{i}$ ([30] considers sufficiently large values when ${\tilde{\gamma }}_{ih}>0.49$), together with estimates of $\beta_{i}$ close to 1/K.

Moreover, it is possible to test if the estimate for $\beta_{i}$ is significantly different from 1/K by constructing an $\chi^{2}$ statistic. In other words, the statistic allows us to test if the estimated ${\tilde{p}}_{im}$ is significantly different from the respective spike prior $q_{im}^{s}$. The Kullback–Leibler divergence measure between the estimated and the a priori probabilities related to the spike prior is:(18)$$D_{i}\left({\tilde{p}}_{i}\|q_{i}^{s}\right)=\overset{M}{\underset{m=1}{\sum }}{\tilde{p}}_{im}ln\left(\frac{{\tilde{p}}_{im}}{q_{im}^{s}}\right)$$

The $\chi^{2}$ divergence between both probabilities distributions is:(19)$$\chi_{M-1}^{2}=M\overset{M}{\underset{m=1}{\sum }}\frac{{\left({\tilde{p}}_{im}-q_{im}^{s}\right)}^{2}}{q_{im}^{s}}$$

A second-order approximation of $D_{h}\left({\tilde{p}}_{h}\|q_{h}^{s}\right)$ is the entropy-ratio statistic for evaluating ${\tilde{p}}_{h}$ versus $q_{h}^{s}$:(20)$$D_{i}\left({\tilde{p}}_{i}\|q_{i}^{s}\right)≅\frac{1}{2}\overset{M}{\underset{m=1}{\sum }}\frac{{\left({\tilde{p}}_{im}-q_{im}^{s}\right)}^{2}}{q_{im}^{s}}$$

Consequently:(21)$$2MD_{i}\left({\tilde{p}}_{i}\|q_{i}^{s}\right)\to \chi_{M-1}^{2}$$

Thus, the measure $2MD_{i}\left({\tilde{p}}_{i}\|q_{i}^{s}\right)$ allows us to test the null hypothesis $H_{0}:\;\beta_{i}=1/K$. If $H_{0}$ is not rejected, we conclude that a predictor $x_{i}$ should be weighted as a simple arithmetic. (We would like to point out that, when computing, log(0) presents problems in the computation. In order to overcome this, in the empirical application on the next section, the spike priors $q_{i}^{u}$ have been specified with a point mass at zero equal to 0.999 and 0.0005 respectively for the other points of the support vectors.) In such a case, the vector of weights of the combined forecast estimated by using the DWP estimator is not different from the simple average. It means that the sample does not contain information providing strong empirical evidence to weigh differently than equal.

## 4. A Numerical Simulation Study

In this section of the paper, we compare the performance of the proposed DWP estimator with other methods used to combine individual forecasts by carrying out a numerical simulation study. Forecast combinations have been successfully applied in several areas of forecasting, such as economy (gross valued added, inflation, or stock returns), meteorology (wind speed, rainfall, see e.g., [47] in *Entropy* journal), or energy fields (wind power), among others. We focus our empirical exercise in the economic area; in fact, we take variable y as the gross value added being forecasted. (It is supposed that y is measured without error. In a situation in which y was measured with error, [48] proposed a method to extend the simple linear measurement error model through the inclusion of a composite indicator by using the GME estimator.)

The starting point of the numerical simulation is the unknown series $y_{t}$ (t=1,…,T) that contains the target variable and a (T×K) matrix X with K potential unbiased forecasters of this series along the T time periods. The basic idea is that X should contain some imperfect information on the target series. Specifically, in the experiment, the elements of X will be generated in the following way:(22)$$x_{it}=y_{t}+u_{it};\;t=1,\ldots ,T;\;i=1,\ldots ,K$$
where $u_{i}\sim N\left(0,\sigma_{i}\right)$ is a noise term that reflects the accuracy of $x_{i}$ as a forecaster of y and $\sigma_{i}$ is a scalar that adjusts the variability of this noise. Note that $\sigma_{i}$ indicates the degree of information for the target series that is contained in predictor $x_{i}$, i.e., the higher the value of $\sigma_{i}$, the less informative $x_{i}$ is about y.

Given that in our numerical experiment we would like to replicate situations normally observed in the context of forecasting economic series, instead of numerically generating the values of our target variable y, we opted for taking actual values of an economic indicator. More specifically, we have taken the annual Gross Value Added rate of change in the region of Catalonia (Spain) from 1980 to 2013. We have extracted this information (at constant prices of 2008) from the BDmores database. (This database is generated by the Spanish Ministry of Economy, Industry and Competitiveness. More details can be found in: http://www.sepg.pap.minhap.gob.es/sitios/sepg/en-GB/Presupuestos/Documentacion/paginas/base0sdatosestudiosregionales.aspx).

Concerning the configuration of matrix X, we consider different numbers of potential predictors (dimension K) to be combined. Given that, in the context of forecasting regional indicators, the number of forecasters is normally smaller than when national or supra-national variables are predicted, we have set three different values for K, with K set to 6, 12, and 24. Moreover, we have considered that the behavior of these predictors can be heterogeneous when aiming at forecasting variable y. In particular, we have divided our set of K forecasters into two different subsets that can be classified as “good” or “bad” predictors. The logic of this idea is that the information that the predictors provide for forecasting variable y can vary among them, with a “good” predictor preferable to a “bad” one, but with the caveat that the comparatively “bad” forecaster may still contain some potentially useful information to be considered in the combination. In order to reflect this idea, the elements of matrix X will be generated differently in the following two subsets:(23)$$x_{it}=y_{t}+u_{it}^{g};\;t=1,\ldots ,T;\;i=1,\ldots ,G$$
(24)$$x_{it}=y_{t}+u_{it}^{b};\;t=1,\ldots ,T;\;i=G+1,\ldots ,K$$
where $u_{it}^{g}$ is the noise term for the subset of G “good” predictors and $u_{it}^{b}$ is the corresponding element for the comparatively “bad” ones. The difference between $u_{it}^{g}$ and $u_{it}^{b}$ is on its variability, since:(25)$$u_{i}^{g}\sim N\left(0,\frac{s}{2}\right)$$
(26)$$u_{i}^{b}\sim N\left(0,s\right)$$
where s is the standard deviation in the sample 1980–2013 of the target variable y. Equation (25) and Equation (26) indicate that the variance of the forecasters classified as “good” presents a variance four times lower than for those classified as “bad”.

In the simulation, we have set different proportions between these two subsets of predictors. First, a more realistic situation where 5/6 of the total of K forecasters belong to the group of “good” predictors and only 1/6 are classified as “bad.” Additionally, and for comparative purposes, a situation where they are distributed in equal parts (50%) to each group is considered as well.

In the experiment, all the simulated predictors are combined through the regression-based method of combining forecasts:(27)$$y_{t}=\overset{K}{\underset{i=1}{\sum }}\beta_{i}x_{it}+e_{it};\;t=1,\ldots ,T$$
with the target of the different methods for combining these forecasters to determine the best possible values for the $\beta^{\prime }s$ parameters.

The benchmark for comparing the competing methods will be the arithmetic mean of the forecasters, where $\beta_{i}=1/K,\;\forall i$, which is normally the strategy taken as a valid reference in the literature on combination of forecasters. In fact, it is sometimes considered as the best way of combining information of individual predictors as some studies have pointed out (for example, [2,10,27,28,29]). Additionally, a restricted least squares weight scheme (see [19], for the original unrestricted Leas Squares approach; or [5] for the restricted version) is considered as well, where the $\beta^{\prime }s$ weights (restricted to sum to one) are estimated by minimizing the sum of squared errors $e_{it}$.

Our comparison is extended to include the proposals made in recent forecasting literature, where forecasts based on Bayesian model averaging (BMA) has received considerable attention (see [49,50]). In this approach, the weights are determined based on the Bayesian information criterion (BIC) as:(28)$$\beta_{i}=\frac{exp\left[-\frac{1}{2}BIC_{i}\right]}{\sum_{i=1}^{K}exp\left[-\frac{1}{2}BIC_{i}\right]};$$
and
(29)$$BIC_{i}=Tln\left({\hat{\sigma }}_{i}^{2}\right)+ln\left(T\right)$$
where ${\hat{\sigma }}_{i}^{2}$ stands for the LS estimation of $\sigma_{i}^{2}$.

These techniques for combining the individual predictors $x_{i}$ will be compared with the estimation of the optimal $\beta^{\prime }s$ weights when the DWP estimator is applied. Consequently, specifying some support for the set of parameters to be estimated and the errors is required. We have fixed the same vector b for all the $\beta^{\prime }s$ parameters. In particular, the proposed DWP estimator assumes as a prior value for each $\beta_{i}$ the solution provided by the simple mean of forecasters, where all are equally weighted as 1/K. More specifically, we have considered that each unknown parameter $\beta_{i}$ has M=3 possible realizations with values $b^{\prime }=\left(1/K-1,1/K,1/K+1\right)$; in other words, the bounds with the minimum and maximum possible values for the weights are set as the center 1/K±1.

For the weighting parameters, we have considered a support vector with two possible realizations N=2 and values $b^{\prime }=\left(0,1\right)$. Finally, the supports of the random error terms have been specified by guarantying symmetry around zero and by using the three-sigma rule (−3s, 0, 3s), with s being the sample standard deviation of the dependent variable.

Table 1 and Table 2 summarize the results of comparing the actual target values of our variable of interest ($y_{t}$) with the combined individual forecasts (${\hat{y}}_{t}$) obtained according to the different methods, namely; the simple mean (mean), Least Squares (LS), Bayesian Information Criterion (BIC) and the proposed Data Weighted Prior (DWP), and following two different deviation measures: (i) The mean squared forecast errors (MSFE); and (ii), the mean absolute percentage forecast error (MAPFE), respectively, defined by the two following expressions:(30)$$MSFE=\overset{F}{\underset{f=1}{\sum }}{\left(y_{f}-{\hat{y}}_{f}\right)}^{2}$$
(31)$$MAPFE=100\overset{F}{\underset{f=1}{\sum }}\left|y_{f}-{\hat{y}}_{f}\right|$$

The mean values of these deviation measures are computed from 1000 trials and for a forecast horizon of four periods ahead (f=1,…,4), which means that the last four periods in our sample are not included in the estimation of the weights, but taken as reference for evaluating the performance of our combination of predictions.

Error figures in Table 1 and Table 2 show how the simple mean outperforms the combining methods based on some regression analysis (LS or BIC) in situations where the number of potential forecasters is large relative to the available sample size. When the predictors considered are 12 or 24, the combination based on LS and BIC presents problems derived from an ill-conditioned dataset (the number of parameters is large relative to the small sample size), whereas the arithmetic mean of predictors is not affected by this problem. The proposed DWP estimator seems to beat the competing combination techniques, given that it takes the weighting scheme as the arithmetic mean and only departs from these weights if the sample contains information providing strong empirical evidence to weigh differently than equal. On the contrary, when the number of predictors is low, an LS-based combination of forecasters performs better than any of the other techniques, given that now the sample size is large enough in relative terms to the number of predictors considered. One important aspect to consider, however, is that the performance of the proposed combined forecast methods has only been evaluated under the criterion of accuracy (measured through some forecast error-based indicators). However, other criteria could be considered (such as forecast error variance or asymmetry) leading to a different relative performance of the combining methods [9].

## 5. Conclusions

One of the most widespread strategies for combining individual forecasts is to take a simple average of the forecasts. Empirically, many studies have shown that the mean outperforms complex combining strategies. Theoretically, the use of the simple arithmetic mean could be justified when all the forecasters have shown the same forecasting ability or when the available information about their ability seems to be not enough to calibrate the forecasters differently. This paper proposes the use of an entropy-based technique estimator to obtain an affine transformation of the equal weighted forecast combination by using the small available information, a data-weighted prior (DWP) estimator.

We tested the validity of the proposed model by a simulation exercise and compared its ex-ante forecasting performance with other combining methods. The benchmarks for comparing the competing method were the arithmetic mean of the forecasters, a restricted least squares, and weight scheme forecasts based on Bayesian model averaging (where the weights are determined on the basis of the Bayesian information criterion).

We set three different values for the number of individual forecasts to be combined (6, 12, and 24) and we have divided our set of forecasters in two different subsets, which can be classified as “good” or “bad” predictors. The obtained results of the simulation indicate that the proposed DWP estimator seems to beat the competing combination techniques, given that it takes the weighting scheme as the arithmetic mean and only departs from these weights if the sample contains information providing strong enough empirical evidence to weigh differently than equal. The most relevant advantage of this estimator is that, even in situations characterized by a large number of forecasters, the DWP estimator generates a better set of recovered forecasters´ weights than the arithmetic mean which is capable to identify groups of forecasters into groups of “good” and “bad” forecasts. Additionally, the empirical application could be extended by comparing the forecasting performance of the proposed method with other combining methods based on an information-theoretic approach [6].

## Figures and Tables

**Table 1 entropy-21-00429-t001:** Mean squared forecasting error (MSFE); 1000 trials.

Mean Squared Forecasting Error (MSFE)					
		Method			
*K*	*G*	mean	LS	BIC	DWP
6	5 good	0.0160	0.0136	0.0298	0.0156
	3 good	0.0269	0.0180	0.0379	0.0261
12	10 good	0.0077	0.0099	0.0256	0.0076
	6 good	0.0128	0.0141	0.0288	0.0125
24	20 good	0.0040	0.0147	0.0191	0.0039
	12 good	0.0064	0.0205	0.0243	0.0062

**Table 2 entropy-21-00429-t002:** Mean absolute percentage forecasting error (MAPFE); 1000 trials.

Mean Absolute Percentage Forecasting Error (MAPFE)					
		Method			
*K*	*G*	mean	LS	BIC	DWP
6	5 good	2.0312	1.8454	2.7303	2.0023
	3 good	2.6217	2.1553	3.1300	2.5799
12	10 good	1.4251	1.5797	2.5231	1.4079
	6 good	1.8182	1.8762	2.7280	1.7976
24	20 good	1.0132	1.8501	2.1976	0.99836
	12 good	1.2749	2.2305	2.5106	1.2556

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
