# Peer review of "A Data-Weighted Prior Estimator for Forecast Combination"

_entropy, 2019, doi:10.3390/e21040429_

Round 1

Reviewer 1 Report

The paper is interesting and present original results in terms of interpretation and application of the Generalized Cross Entropy method. The introduction of the objective function Eq. (10) is well-motivated and the comparison with the existing techniques for combinations of forecasters (Introduction, and pp. 5-6) are adequately discussed. The set of simulations are neatly and clearly analyzed. 

In order to improve the exposition of the results, some remarks about minor clarifications and amendments are reported in the attached file. Moreover, please, upload the references with the following papers: 

- Angelelli, M., Ciavolino, E. (2018). Streaming Generalized Cross Entropy. arXiv preprint arXiv:1811.09710.

- Carpita, M., Ciavolino, E. (2017). A Generalized Maximum Entropy Estimator to Simple Linear Measurement Error Model with a Composite Indicator. Advance in Data Analysis and Classification. 11.1 (2017): 139-158.

Reviewer 2 Report

This paper presents a study to combine multiple predictions to a single predictor. The paper is not recommended for publication due to the following reasons:

 Section 1 presents the introduction and literature. Section 2 introduces the problem to be investigated. Section 3 presents the Data-Weighted Prior (DWP) Estimator, which is an existing method while Section 4 presents a numerical study. The method is an existing method, namely Data-Weighted Prior (DWP) Estimator. I am unable to find further development or novelty of the paper.

There are a number of language problems or misuse of terms.

The presentation needs to be improved too as some of the symbols were not defined.

Author Response

The authors would like to thank you very much for the careful reading of our work and the useful suggestions proposed in your report.

According to your suggestions (in blue) we have corrected our manuscript. Following, we explain the changes made point by point. We hope the new version of the manuscript properly addresses them. In any case, we would be happy to make any other changes you consider necessary.

Comment#1: I am unable to find further development or novelty of the paper.

The novelty of the paper is to extend the use of the Data-Weighted Prior (DWP) Estimator proposed by Golan (2001) as a procedure to select forecasters and simultaneously to estimate the weights associated to each forecast in the combined forecast. The novelty is not the estimator itself, but the development of a procedure to use it in the forecast combination framework in a situation of ill-conditioned datasets.

We acknowledge this right from the beginning. In the abstract we state that “(…)With this purpose, the Data-Weighted Prior (DWP) Estimator proposed by Golan (2001) is used for forecasters selection and simultaneous parameter estimation in linear statistical models.” We later inset on this at the end of the introductory section by explaining that “(…) under regression-based combination method framework we propose a procedure that allows for simultaneous parameter estimation and forecast selection in linear statistical models. The procedure is based on the Data-Weighted Prior (DWP) estimator proposed by [30].” We have highlighted this by modifying the introductory section.

The novelty of the paper could be summarized as follows: There are several methods to estimate the weights of the individual forecast to be combined ([8]-[19], among others) and several procedures to take out relevant information from a large number of forecasts ([20]-[26]). Sometimes there is not available information about individual forecast´s past performance to calibrate differently them or the number of institutions producing forecasts is large and the calculation of the combined forecast would involve the estimation of a large number of weights. In such situations, it is usual to use the simple mean average of the individual forecasts as a combined forecast. In these contexts, we analyze how DWP is able to reduce the number of potential forecasters and simultaneously to estimate a vector of weights different from the simple average in the combined forecast, even in that situation of small information.

Comment#2: There are a number of language problems or misuse of terms.

According to your suggestion we have the revised version the English language of the manuscript to eliminate possible grammatical and spelling errors.

Comment#3: The presentation needs to be improved too as some of the symbols were not defined.

We have defined some symbols as  , n or  , among others

Round 2

Reviewer 2 Report

The authors have addressed all my previous comments and the paper is recommended for publication.